# Trans Abroad: American Transgender Students' Experiences of Navigating Identity and Community While Studying Abroad

Taylor Michl [1,*] , Alexandra Stookey [2], Jillian Wilson [3], Katie Chiou [4], Trisha L. Raque [5] and Amanda Kracen [6]

1 Department of Counseling and Clinical Psychology, Teachers College, Columbia University, 525 W 120th St., New York, NY 10027, USA
2 College of Health Solutions, Arizona State University, Phoenix, AZ 85004, USA; astookey@asu.edu
3 Department of Psychiatry, Washington University in St. Louis, St. Louis, MO 63130, USA; jillianw@wustl.edu
4 Warren Alpert Medical School, Brown University, Providence, RI 02903, USA; katie_chiou@brown.edu
5 Morgridge College of Education, University of Denver, Denver, CO 80210, USA; trisha.raque@du.edu
6 Department of Psychology, School of Business, National College of Ireland, Dublin 1, Ireland; amanda.kracen@ncirl.ie
* Correspondence: tm3351@tc.columbia.edu

**Abstract:** Despite significant and increasing numbers of students studying internationally, there are few data about the experiences of study abroad for various marginalized students, including transgender and gender expansive (TGE) students. Therefore, 15 TGE adults from the United States were interviewed about navigating gender and culture during undergraduate study abroad programs. Interviews were analyzed using consensual qualitative research (CQR). Participants shared how they benefited from international study and navigated intersecting social identities, including gender, which was complex and nuanced. They discussed how they actively managed issues of disclosure and its consequences, explored their identities and the influence of their social setting, and dealt with anticipated, deliberate, and unintentional harm from others. Relationships and community were priorities for participants when studying internationally; they explained how they determined whether to invest in relationships or not. Participants also clarified what their relationships looked like during study abroad, as well as unique considerations that arose from their marginalized identities. These findings can help inform the development of more inclusive, safe, and satisfying study abroad experiences for all students, especially TGE individuals; implications for future research and study abroad interventions are provided.

**Keywords:** transgender; gender expansive; study abroad; identity; community; minority stress; consensual qualitative research; intersectionality





## 1. Introduction

Studying abroad can be students' first time traveling internationally without family members. It is a unique time to explore their world, their beliefs, and themselves. As identities are socially and culturally constructed (Jackson and Hogg 2010), studying internationally offers students a rare opportunity to examine who they are against the backdrop of a new culture. Students, including transgender and gender expansive (TGE) people, can experience and perform gender in ways that may be new to them.

TGE people have gender identities that do not align with their sex assigned at birth (NCTE 2023). Although TGE people should be valued and celebrated members of society, they often experience deleterious responses from others. In fact, the wellbeing of TGE individuals is currently under attack, as targeted hate crimes have been on the rise and many U.S. states are pursuing anti-LGBTQ+ legislation. Sadly, 19% of all reported hate crimes in the U.S. were targeted towards LGBTQIA+ communities in 2021 (FBI 2021), and at least 34 transgender and gender expansive individuals were murdered in the U.S. in 2022 (HRC 2022). According to the American Civil Liberties Union (ACLU 2023), there

are currently 492 active anti-LGBTQIA+ bills in the U.S; it is widely recognized that such legislation causes harm.

According to Meyer (2003), individuals in minoritized groups experience minority stress, or unique distress because of how socially marginalized identities are treated in society. Minority stressors range from distal (external) to proximal (internal) stressors. Common distal TGE stressors include violence, rejection, lack of access to legal documents and medical care, inability to access safe bathrooms, and non-affirmation or misgendering (Testa et al. 2015). The U.S. Trans Survey (USTS; James et al. 2016), the largest survey exploring U.S. TGE experiences, found that existing in public as a TGE person in the U.S. can be incredibly dangerous; 48% of over 27,000 survey participants reported experiencing discrimination, harassment, and/or physical attacks in the previous year related to their TGE identities. Spaces such as public transportation, airport security, and restrooms can be especially high-risk. In addition, while colleges and universities are often settings that foster personal growth, safety, and community, they can simultaneously be places for anti-TGE violence. The USTS found that 24% of survey participants who were out as transgender at their college or university experienced verbal, physical, and/or sexual harassment or violence at school. Rates of discrimination, violence, and mistreatment in all settings are even higher for TGE students of color.

As TGE individuals living in the U.S. regularly receive harmful cultural messaging that their identities are pathological, that they will be subjected to harm, and that they are not welcome in public spaces, it is understandably difficult to resist internalizing these ubiquitous messages. Proximal stressors, which are often experienced in response to distal stressors, include internalized transphobia; fear of future discrimination, rejection, and violence; identity concealment; and gender dysphoria (Testa et al. 2015; Lindley and Galupo 2020). The internalization of distal stressors, which results in proximal stressors, can limit TGE individuals' access to resilience factors that buffer the negative influence of minority stress on mental health outcomes. For example, if a TGE person is actively concealing their identity to maintain their safety, they may not reach out to TGE community members. Additionally, TGE people with internalized transphobic beliefs will likely not experience identity pride. Experiencing both distal and proximal minority stressors can lead to adverse outcomes for mental health and wellbeing, including psychological distress and suicide.

*1.1. Transgender Identity Development*

In addition to minority stress factors, there are factors that can buffer the negative impacts of minority stressors and promote resilience for TGE folks, including identity acceptance and pride; and community connectedness, solidarity, and cohesiveness (Meyer 2003; Testa et al. 2015). Prior to experiencing pride and connection with the TGE community, TGE people often undergo a complex process of identity development. According to the USTS, 60% of participants reported that they first felt different from their sex assigned at birth before age 10 (James et al. 2016). A majority of participants first came out as TGE between the ages of 16 and 30 and began to socially, physically, and/or medically transition between 18 and 34. Research also shows that key steps in gender transition often occur during emerging adulthood (Kuper et al. 2018). As study abroad often occurs during emerging adulthood, TGE study abroad students may experience critical components of their identity realization, exploration, and transition processes while abroad.

According to Doyle (2023), the TGE identity development process often, but not always, includes the following milestones: childhood gender nonconformity, personal feelings of gender incongruence, gender dysphoria, self-awareness of gender identity, and public self-identification as TGE. Whereas disclosing a TGE identity can lead to increased psychological wellbeing, affirmation from others, and community solidarity, coming out can also pose risks to TGE individuals' safety and wellbeing due to the distal minority stressors they may face. It is important to note that both the processes of identity development and identity disclosure can be continuous and nonlinear for TGE people; coming out is often a repetitive process involving negotiating the unique trust, boundaries, and safety

considerations within each relationship. In fact, the USTS revealed that it is relatively uncommon for TGE people to have disclosed their gender identities to absolutely everyone or absolutely no one in their lives (James et al. 2016). Instead, a majority of participants were either out to "some" or "most" people in their lives.

As gender is constructed, the process of understanding one's own gender identity occurs through interactions with and comparisons to the experiences of others and the master narratives in society (Kuper et al. 2018). Study abroad offers the opportunity to explore and develop identity by exposing students to a new cultural context and inviting them to integrate these experiences into existing understandings of themselves. Bradford and Syed (2019) explore master and alternative narratives impacting the development of gendered beliefs, attitudes, and biases. Cisnormativity is a master narrative operating in many societies that assumes everyone is and should be cisgender or have a gender identity that aligns with their sex assigned at birth. Transnormativity is an alternative narrative that normalizes and validates the existence of some transgender people (e.g., those who medically transition, identify within the gender binary, enact normative gender roles, have known they were transgender since childhood, and experience victimization and tragedy related to their trans identities), while positioning others as illegitimate or "not trans enough." In addition to the internal and intrapersonal aspects of identity development discussed earlier, TGE identity development also involves contending with both cis- and transnormativity in order to create personal gender narratives that either comply with or resist these metanarratives.

As TGE communities are diverse in their racial, ethnic, disability, class, sexuality, and other identities, it is important to attend to how gender is constructed in tandem with other social identities. Intersectionality (Crenshaw 1989) refers to the relationships between an individual's multiple social identities and their corresponding social systems. Crenshaw (1989) argues that, rather than stacking on top of one another, social identities intersect to create unique, complex, and multifaceted human experiences. For example, while a White transfeminine person's experience may be impacted both by gender and race, including systems of cisnormativity and White supremacy, these identities also intersect to create an entirely unique lived experience. Their experiences of race, gender, and their other identities are not merely additive; instead, their White racial identity is constructed alongside and shaped by their transfeminine identity and vice versa. Every human being experiences a complex interplay between their socially privileged and marginalized identities and the social and cultural systems in their environment.

### 1.2. LGBTQIA+ Study Abroad Research

For the current research project, study abroad refers to international education opportunities that U.S.-based students take part in at the undergraduate level. To date, research on TGE students' experiences during study abroad is incredibly scarce. In their study on LGBTQIA+ students' participation in study abroad programs, Bryant and Soria (2015) found that LGBTQIA+ students may be more likely than their straight and cisgender peers to participate in study abroad opportunities. However, Hipple et al. (2020) found that many colleges and universities neglect to include LGBTQIA+ communities in demographic-specific study abroad outreach. When LGBTQIA+ students are explicitly incorporated in study abroad outreach and other resources, Siddiqui and Jessup-Anger (2020, p. 460) reported that TGE students are often only "superficially included", but their specific needs are not addressed.

Many have recommended that more research on the safety, identity disclosure, intersectional identity, community, and sexual violence experiences of LGBTQIA+, and specifically TGE, study abroad students be conducted (Bryant and Soria 2015; Siddiqui and Jessup-Anger 2020; Bingham et al. 2023). Young et al. (2015) encouraged that identity development be explored with students who are studying abroad in a wide variety of international locations and durations. Hipple et al. (2020) specifically recommended future

explorations centered on the interests and concerns of LGBTQIA+ students (including TGE students) studying abroad.

Two members of the current research team and a former colleague previously completed a pilot study exploring the study abroad experiences of three TGE students (Michl et al. 2019). Although the sample was small, the results were rich, indicating that the students were actively engaged in reflecting on their gender and other identities when navigating a new culture. Participants discussed experiencing growth and shifts in gender identity and expression as a result of study abroad. They shared how they learned what was and was not acceptable in terms of performing gender in their host culture and their experiences of being influenced and policed by others when it came to gender expression. In terms of community, these participants expressed the importance of queer community and chosen family abroad because the bonds were fulfilling and special. The pilot study also found that participants experienced a lack of preparation and institutional support. Unfortunately, all three participants had experiences of sexual harassment and/or violence while studying abroad and feared potential future violence. The findings of the pilot study clarified that students' experiences were so rich that a follow-up study was warranted, which inspired this study.

### 1.3. Study Aims

The current study expands on the findings from the pilot study by using a larger sample size and more robust methodology and clarifies how study abroad is experienced by TGE individuals in order to address gaps in the existing research literature. This study explored the following research questions: (1) How do TGE individuals make sense of their undergraduate study abroad experiences? (2) How do TGE individuals experience gender and culture in the context of study abroad? As cultural values affect the treatment of gender minority communities in different countries, this research both highlights TGE students' experiences and yields insights about cultural differences. Rather than focusing specifically on one location, this research seeks to understand the study abroad experiences of TGE students more generally.

## 2. Materials and Methods

This was a qualitative research study. Demographic surveys and semi-structured phone and Zoom audio interviews were completed for 15 participants during the summer of 2020. Consensual qualitative research (CQR; Hill et al. 1997, 2005) was used to analyze the data. CQR's philosophical underpinnings are constructivist and postpositivist in nature (Hill and Knox 2021). Experiences of identity, personal growth, and community during study abroad can be complex, internal, and not easily observed from an outsider's perspective; CQR was used because it is a rigorous qualitative methodology that allows researchers to capture participants' subjective ways of seeing the world. CQR is particularly useful for exploratory inquiry of topics that have not been extensively studied (Hill and Knox 2021). Additionally, CQR acknowledges the inevitability of researcher biases, invites open conversation about these biases, and uses multiple perspectives, consensus, and auditing to bracket biases and minimize their impact on data collection and analysis (Hill and Knox 2021). Given that our research team was comprised of several members with experience studying and teaching abroad and who identify as TGE and/or LGBQ, we thought it was critical to utilize a qualitative approach that structurally addressed the potential influence of researcher bias, noticing it and making it more transparent in our coding.

### 2.1. Researcher Description

At the time of data collection, the research team was composed of five individuals from several universities, ranging in age from 20 to 44, two identifying as gender expansive and the others identifying as cisgender women; three self-identifying as White, one as Black, and one as Asian American. One member of the research team (White, nonbinary)

parted ways in early 2021 due to scheduling conflicts, and another joined for the remainder of the project (White, cisgender woman). The principal investigator (first author) was a graduate student, mentored by a university faculty member (last author). Three team members were undergraduate students in psychology and anthropology. Multiple team members' university affiliations changed throughout the data analysis. All research team members participated in training about conducting research with TGE populations, led by the first author, and a CQR training led by the fifth author, who served as the project's auditor and is an expert in CQR.

The research team formally and informally discussed their lived experiences and possible biases at the start of the research project and during subsequent research meetings. Team members shared that numerous experiences may impact their views on the data, including their involvement in conducting a pilot study; their gender and racial identities; their own experiences of studying and teaching abroad; the power dynamics on the research team related to age, racial identity, lived experience, and professional experience; and current and former dual relationships as professors and students. These conversations allowed team members to transparently acknowledge the ways their own and others' experiences may influence their views of the interview data and codes. Along with these conversations, researchers' lived experiences and biases were bracketed during the data analysis processes of achieving consensus and auditing.

### 2.2. Ethical Considerations

The Institutional Review Board at Webster University granted approval for the current study. Due to the history of exploitation that TGE and other marginalized groups have experienced at the hands of researchers, we were committed to thoughtful and reflective research processes. We engaged in dialogue and consulted resources related to ethical issues in conducting research with TGE participants throughout the research process. This approach was taken in the hopes that the data collection, analysis, and dissemination processes would contribute to the thoughtful and accurate depiction of diverse and intersectional TGE stories, rather than further contributing to the historically harmful and othering relationship between researchers and TGE participants (Meezan and Martin 2009; Krumer-Nevo 2012). As such, the research was led by a gender expansive researcher (first author), and this was communicated to potential participants on recruitment materials to enhance safety. Written informed consent was obtained through an online survey and verbal informed consent was confirmed prior to the interview. Additionally, it was important to us that participants were fairly compensated for the time, energy, and vulnerability asked of them. Participants were compensated in the form of purchases from an online retailer of their choice valued at $40.

### 2.3. Study Participants

2.3.1. Participants

Participants ($N$ = 15) ranged in age from 20 to 29 and identified as transmasculine/transmen, nonbinary/gender nonconforming, and transfeminine/transwomen. Participants' racial/ethnic identities included multiracial, White, Black/African American, Latine/x, and Asian/Pacific Islander. Participants studied abroad in Europe, North America, South America, Africa, and Asia. More detailed participant demographics are displayed in Table 1; to protect participants' anonymity, demographics are reported for the group.

**Table 1.** Participants' self-reported demographics.

| Demographic Category | Participant Demographics | *N* = 15 |
|---|---|---|
| Gender | Nonbinary | 5 |
| | Female | 2 |
| | Male | 1 |
| | Nonbinary genderfluid | 1 |
| | Trans femme | 1 |
| | Genderqueer | 1 |
| | Genderfluid | 1 |
| | Transmasculine/Nonbinary | 1 |
| | Transmasculine/GNC | 1 |
| | Black nonbinary person | 1 |
| Study Abroad Location | Europe | 9 |
| | North America | 2 |
| | South America | 2 |
| | Africa | 1 |
| | Asia | 1 |
| Race/Ethnicity | White | 4 |
| | Black or African American | 3 |
| | Latinx | 3 |
| | Latinx and Native American or Alaska Native | 1 |
| | Latinx and White | 1 |
| | Native American or Alaska Native and Chicanx | 1 |
| | Asian and White and Hispanic | 1 |
| | Asian/Pacific Islander | 1 |
| Sexuality | Queer | 3 |
| | Bisexual | 3 |
| | Pansexual/Queer | 2 |
| | Demisexual | 1 |
| | Straight | 1 |
| | Lesbian/Queer | 1 |
| | Asexual Polyromantic | 1 |
| | Queer/Gay | 1 |
| | Lesbian | 1 |
| | Gay | 1 |
| Age | 20–24 | 13 |
| | 25–29 | 2 |
| Education Level | Current undergraduate student | 3 |
| | Two-year college diploma completed | 1 |
| | Bachelor's degree completed | 10 |
| | Current master's student | 1 |

### 2.3.2. Researcher–Participant Relationship

Participants interacted with the first author when volunteering to participate in the study, scheduling the interview, completing the interview, and coordinating the participant honoraria. Two participants were briefly acquainted with the first author in group settings prior to initiating the research study. However, these prior interactions did not explicitly impact the data collection or analysis processes.

### 2.4. Recruitment and Participant Selection

Participants were recruited in the United States through purposeful and snowball sampling. The study announcement was (1) posted to Facebook groups dedicated to LGBTQIA+ individuals and the personal Facebook, LinkedIn, and Instagram profiles of the researchers and (2) emailed to study abroad department administrators, social science professors, and colleagues at universities. To participate, individuals had to (a) be over the age of 18 years, (b) be a citizen or resident of the United States, (c) have studied abroad outside of the U.S. in the past two years, and (d) be transgender and/or gender expansive.

Interested participants were directed to an online Qualtrics link to learn about the study, provide informed consent and contact information, and provide demographic details. After participants completed an interview, they were invited to share information about the study with others who may be interested. It is recommended that CQR studies use 13–15 participants to yield consistency across participant narratives and achieve saturation (Hill and Knox 2021). Eighteen participants met all inclusion criteria, provided consent, and agreed to be interviewed. Of those, 15 completed the interviews; two did not due to scheduling conflicts and one volunteered after the study was closed.

*2.5. Data Collection*

Data collection occurred during the summer of 2020, when many were impacted by the COVID-19 pandemic and anti-Black police violence in the U.S. While these events were not a focus of the present study, researchers acknowledged the pain and exhaustion associated with them during recruitment and data collection so as to honor the lived experiences of intersectional participants.

2.5.1. Data Collection Procedures

The data were collected in the form of demographics questions and a semi-structured interview. The demographics questions, in an online questionnaire, asked participants their age, student status, which college/university they attend(ed), highest level of education, number of times they had studied abroad, study abroad location, study abroad timeline, living situation abroad, gender identity, sex identity (i.e., intersex or endosex), pronouns, sexuality, relationship status and structure, and racial/ethnic background.

Interviews were conducted by the first author using a semi-structured protocol of 14 questions and several possible prompts (see Appendix A). The interview protocol was developed after reviewing the scarce literature available on the study abroad experiences of LGBTQIA+ individuals, including findings from the pilot study (Michl et al. 2019). Interview questions explored motivations and preparation for study abroad; identity, relational, safety, and personal growth experiences during study abroad; and advice that participants would give future TGE study abroad students. The draft interview protocol was piloted with two TGE individuals who had studied abroad and minor adjustments were made. Once finalized, participants were provided with the interview protocol via email to familiarize themselves with the topics before the interviews. All interviews were conducted either over the phone or via Zoom with the cameras off as recommended by CQR guidelines to allow for greater participant anonymity when disclosing sensitive information related to marginalized identities (Hill and Knox 2021). While all participants were asked the same open-ended interview questions, the interviewer prompted participants to elaborate and asked individualized follow-up questions as necessary. The interview duration ranged from 27 to 60 min, with an average interview time of 47 min.

2.5.2. Recording and Data Transformation

All interviews were recorded using an external audio recording device. The first and fourth authors manually transcribed the interviews; the transcripts were sent to participants to review and amend any responses as they deemed necessary. Two participants suggested minor clarifications, which were made to their transcripts.

*2.6. Analysis*

2.6.1. Methodological Integrity

CQR requires the use of an auditor who is situated outside of the research team to serve as an external check. The auditor of this project was an associate professor in a counseling psychology graduate program with expertise in the CQR method and LGBTQIA+ research. The auditor's outsider stance allowed the research team to confirm the accuracy of their analysis and consider a perspective that had not been impacted by the group process of data analysis. The auditor's feedback at the level of domains, core ideas, and cross analysis was

reviewed by the research team, discussed until consensus was reached, and incorporated into the final analysis.

2.6.2. Data Analytic Strategies

The data were analyzed using the inductive CQR method. Apart from the fifth author, who served as the auditor, all authors participated in each step of the CQR process during weekly group meetings and tasks completed between meetings. Coding only took place if there was a minimum of three authors present, although typically there were four to five authors in attendance. It is worth noting that a strength of CQR is the rigor of the research team attending to its biases; our research team consistently prioritized the collaborative, collegial, and challenging nature of CQR. As these data were complex and nuanced, we sought to develop trust in questioning our assumptions and coding decisions. We strove for a flat hierarchy in meetings, took turns carrying out coding tasks, discussed biases, checked in frequently to challenge our interpretations, reviewed interview transcripts to assess the accuracy of our interpretations, and encouraged dissenting opinions.

After reviewing the transcripts, the research team created a preliminary list of domains. The preliminary domains were applied to all transcripts and refined until a final list was constructed and reviewed by the auditor. Core ideas were developed, then reviewed by the auditor to ensure accuracy and clarity. Finally, the team completed cross analysis by developing categories and subcategories relevant to each domain using core ideas. These categories and subcategories were coded onto the data to confirm that they were both relevant and encompassing. Whereas each participant shared unique experiences, several patterns and themes emerged throughout the data analysis. In line with Hill and Knox (2021), the majority of new categories and subcategories emerged while analyzing the first two thirds of cases, indicating that saturation had been reached. The final categories and subcategories were reviewed by the auditor. Qualitative software was not used to organize the data, as is standard in CQR; this approach allowed the researchers to remain close to the data throughout the duration of analysis.

CQR requires that the frequencies of domains, categories, and subcategories be reported to demonstrate to what extent the themes represent the experiences of the sample (Hill and Knox 2021). Domains, categories, and subcategories that are endorsed by 14 or 15 participants are labeled General; 8–13 participants are labeled Typical; 2–7 participants are labeled Variant; and those endorsed by one participant are not reported. According to Hill and Knox (2021), the emergence of General and Typical categories is an additional indication that saturation has been reached.

## 3. Results

The data analysis resulted in seven domains with multiple categories and subcategories within each domain. In this paper, we are presenting two of the seven domains from the broader study, as they are useful in understanding how gender and social identities are constructed and experienced during study abroad. These two domains are: (1) reflecting on identity and personal growth and (2) developing relationships, discerning community, and interacting with others.

### 3.1. Reflecting on Identity and Personal Growth Domain

This domain includes participants' dynamic processes during study abroad of experiencing and reflecting on their various identities and others' reactions. Participants discussed making decisions about identity disclosure; feeling (in)visible, (dis)empowered, and (un)safe in their identities; and shifting identity salience. They shared how they navigated experiencing and expressing gender and other intersectional identities. For example, a self-described fat, transfeminine, Latinx participant reported that sometimes, when pursuing romantic or sexual encounters with queer men, they experienced intersectional discrimination in the forms of fatphobia, femme-phobia, and transphobia. Sometimes, this

led them to change their gender presentation so that they appeared more masculine to preserve their energy and protect their safety in these spaces.

Participants also shared intrapersonal reflections and instances of personal growth resulting from studying in another country. For instance, one participant shared that their main growth area as a result of study abroad was the ability to accept their identity. They discussed that, because they were able to explore their identity abroad "without any of the preconceived notions of who [they were]," they returned to the U.S. a more authentic, outgoing, and stronger version of themselves. Five categories of themes were discerned in the data. See Table 2 for a list of categories, subcategories, and frequencies.

**Table 2.** Categories and subcategories within the Reflecting on Identity and Personal Growth domain.

| Categories (Frequencies) | Subcategories (Frequencies) |
|---|---|
| Making identity disclosure decisions (General, 15 participants endorsed) | Interactions between (in)visibility and agency in disclosure decisions. (General, 14 participants endorsed)<br>Trust associated with disclosure. (General, 14)<br>Novel challenges associated with disclosure abroad. (Variant, 7) |
| Context elicits identity salience (General, 15) | Identities voiced by participants. (General, 15)<br>Situational visibility facilitates awareness and prominence. (General, 15)<br>Privilege affects identity salience. (Variant, 7) |
| Experiencing identity change and personal growth from studying abroad (General, 15) | No subcategories. |
| Enduring harm and anguish related to participants' bodies and identities (General, 14) | Bearing the weight of anticipated harm. (Typical, 12)<br>Experiencing deliberate and unintentional harm. (Typical, 11)<br>Experiencing gender dysphoria. (Variant, 2) |
| Consequences of identity disclosure decisions (Typical, 12) | Positive impacts of disclosing. (Typical, 9)<br>Negative impacts of disclosing. (Variant, 7)<br>Negative impacts of not disclosing. (Variant, 4)<br>Positive impacts of not disclosing. (Variant, 2) |

3.1.1. Category: Making Identity Disclosure Decisions

In this general category, participants described their process of deciding when, whether, how, and to whom they disclosed their personal identities; they discussed how they shared information about their gender, sexuality, disability, and ethnicity. Generally, participants discussed physical expression and the visibility of their identities. While some participants felt empowered to make disclosure decisions, others did not experience autonomy in the disclosure decision-making process due to their physical appearance and other factors. For many, the more visible their identities were, the less agency they had in making selective and intentional disclosure decisions. Sometimes, knowing that others could "see" their identities was comforting and affirming. Other times, this unintentional disclosure resulted in participants worrying about being harmed by others. The way others perceived participants' identities based on their physical expression often depended on the cultural and gender norms of the location. For example, a participant reported that what constitutes normative masculinity in Chile aligns with a more feminine, "metrosexual" version of masculinity in the U.S. Another participant shared that, while their queer sexuality was visible, their gender identity was not, allowing them to make intentional decisions about with whom to share their trans identity while abroad. They explained:

> *I'm definitely myself but I'm not walking down the street telling people I'm trans... I look really gay. And it's not something I can fully help right now, but I'm not a clock-able trans person... I think people think I'm a butch lesbian or something even though that's*

*really not the case, so I think that's the way people look at me when I move. So unless I'm disclosing it, there is no disclosure on my body most of the time. Nobody's walking around looking at me like I'm a trans, genderfluid person . . . that's not something they're able to see.* (Participant 8)

Participants generally reported that trust was an important factor regarding disclosure of important aspects of their identity. Participants described assessing the level of trust and safety that existed within a given relationship, which informed their decisions of whether, when, and how to disclose. Participants shared that in academic environments with supportive professors or social environments with understanding peers, for example, there was significant interpersonal trust and safety, which led them to openly discuss their gender, sexuality, and/or disability identities. Another participant shared how they were more discerning in their disclosure decisions. Although they felt safe disclosing to their host mother that they were gay because this was "consumable" to her, they decided not to disclose their non-binary gender identity to her because they were unsure of how she would respond and whether she would "get it".

Participants variantly described novel challenges of disclosing their identities abroad that they did not deal with in the U.S. For instance, one participant discussed that they had to adjust to speaking a new language in their host country. Once they felt more comfortable navigating the language, they felt more at ease to open up to their friends abroad about their non-binary identity. In addition to language barriers, participants noted that understanding cultural nuances created hurdles in making disclosure decisions. One participant, for example, indicated they did not want to "set [themself] further apart", particularly because they felt their safety hinged on being able to blend in with a group. Contending with others' perspectives of their identities in their host country was also challenging and affected disclosure decisions; for instance, one participant felt they had to emphasize that their experiences as a Black American nonbinary lesbian were not equivalent to those represented by Western media.

### 3.1.2. Category: Context Elicits Identity Salience

In this general category, all participants described their various identities and the relationships among their intersecting identities, especially related to the wider social context. For instance, a participant described that their gender and sexual identities were inextricably linked and equally salient during study abroad:

*Everyone knew that [I was gay] instantaneously. And the non-binary identity was essentially intertwined with that. At least, in my mind, the way my queer identity exists, it's completely one thing. It just has different components. And, for me, I'm always, whenever I'm expressing my queer identity, I'm expressing both the, "I'm gay" and "I'm non-binary." These are both things that make up this queer person that I am. . . That's the identity that I'm most comfortable with and most confident in . . . I made it very prominent that I was a queer person regularly.* (Participant 14)

Identity salience was also indirectly communicated based on which personal identities participants chose to voice during the interviews. They shared many aspects of their demographic identities, including gender identity, gender expression, race/ethnicity, skin tone, disability status, chronic illnesses, class, first generation status, native language, body size, and which U.S. state they were from. Participants also described other aspects of how they saw themselves during interviews, using words like passionate, introverted, homebody, colorful, and creative.

Participants generally discussed that the prominence of their identities shifted based on how visible they were and other external factors, such as the environment, the activities they were doing, and the people they were around. For example, a participant described that their disability, ethnic, and gender identities felt salient during study abroad:

*I think the most prominent for me was being legally blind because that basically affects how I navigate the world and being able to have access to transportation was amazing. So*

*that was really awesome . . . being Mexican American and being trans and legally blind were so at the forefront all the time because I was in student politics. . . But I think that being trans was . . . actually pretty big for me because it kind of would come up a lot with my trans friends who I hung out with a lot. And then being international would actually get brought up a lot, so.* (Participant 3)

Another participant shared that the salience of their racial, gender, and sexual identities shifted significantly while abroad due to studying in a majority Black country. They noted that, as a Black person in America, they are used to being a racial minority, and they are most aware of their Blackness while in the U.S., as White supremacy causes them to "question [their] physical safety." However, in their host country, their Blackness allowed them to "blend in," and people with whom they interacted paid far more attention to their lesbian and non-binary identities than their racial identity. While abroad, this participant felt othered on the basis of their gender and sexual identities and included on the basis of their racial identity.

Variantly, participants discussed how their identities were also impacted by their experiences of privilege compared to others in their immediate social environment. Participants described experiencing racial/ethnic and skin color privilege abroad, some of which were similar to their experiences in the U.S. (especially for White participants), while others reported being part of "a group in power" for the first time while abroad due to differences in racial and ethnic dynamics between the U.S. and host countries. Along with race/ethnicity and skin color, participants noted the privilege of "having greater access" to their host cultures than other study abroad students did because of their fluency in the host language. They also discussed the impacts of experiencing class privilege abroad, including feeling safer to express their gender authentically and being able to study abroad in the first place. Finally, one participant gained clarity about the privilege they experience in the U.S.; because their host culture was not as accepting of gender diversity, they realized that they had taken for granted the opportunity to embody their identity in the U.S.

3.1.3. Category: Experiencing Identity Change and Personal Growth from Studying Abroad

This category, which was generally endorsed, includes participants' perspectives on how they grew as a person from their experience of study abroad. For some participants, experiencing identity change and personal growth felt voluntary; for others, it was experienced as mandatory in order to adapt to their environments. Many participants described feeling like they had "way more tools in [their] pocket" when it came to advocating for themselves as TGE people and expressing their identities after studying abroad. One participant shared that study abroad allowed them to realize that anti-Black and transphobic violence is everywhere. While this is an unfortunate reality, this realization was freeing in a way, as it allowed the participant to feel motivated to travel more, saying, "the violence is inescapable but I can have options".

This category also reflects the holistic and intersectional nature of personal growth that results from study abroad experiences; while some of the growth that participants experienced was related to their gender and other identities, many examples explored aspects of personal development that are unrelated to gender. Participants noted a variety of growth experiences; for instance, they became more independent, observant, understanding, self-efficacious, interpersonally aware, environmentally conscious, and resilient. Some participants described a process of recognizing their own unmet needs and making the changes required to fill these gaps. Others described that traveling to uncomfortable environments allowed them to realize what they appreciated about living in the U.S.; one explained:

*While being in the U.S., I was taking things for granted because . . . the people I usually hang out with are really similar to me, but stepping out of that comfort zone and going . . . abroad to a community that's different in values and beliefs but similar in skin color, it's strange. But it's made me think, you know, basically not take things for granted how I lived in the States.* (Participant 15)

### 3.1.4. Category: Enduring Harm and Anguish Related to Participants' Bodies and Identities

In this general category, participants experienced and anticipated harm in reference to how others perceived their bodies and identities, particularly related to gender. Anticipating harm led participants to carry the emotional weight of being a target; participants discussed feeling unsafe, vulnerable, hypervigilant, and scared. They were appropriately fearful of being physically, sexually, and psychologically harmed due to their marginalized identities. Bearing the weight of this anticipated harm caused many participants to feel limited in their ability to safely engage in certain activities during study abroad, such as being emotionally intimate with romantic/sexual partners, traveling alone, going out to nightclubs and bars, wearing affirming clothing, and using public restrooms. One participant shared that study abroad may have felt more freeing if they were cisgender, as much of their energy was spent wondering how they were being perceived and protecting themselves.

Typically, participants shared experiences of enduring identity- and body-related harm while studying abroad. Whether the harm was deliberate or unintentional, its impacts were damaging. Participants experienced a wide variety of identity- and body-based harm, including being sexually assaulted, targeted, fetishized, misgendered, rejected, excluded, discriminated against, tokenized, and infantilized. Participants also discussed instances of others "challenging" their gender identities, harassing them, assuming their genders, making racist comments, and asking invasive questions about sex and gender. One participant described experiencing the harm of being misgendered, which resulted in feeling isolated:

> *So, [gender] definitely affected my experiences, and I think I also was feeling really vulnerable with my, just not passing . . . because my partner is trans but he was further into his medical transition than I was, and so I did feel these moments of, " . . . everyone is able to have so much fun and feel so free tonight but I was misgendered an hour ago and no one knows what that feels like. . ."* (Participant 11)

Variantly, only two participants reflected on their personal experiences of distress related to experiencing and attempting to cope with gender dysphoria during study abroad. Gender dysphoria is used here to describe participants' personal experiences of distress related to gender misalignment, rather than describing a psychological diagnosis. For example, a participant shared that they felt uncomfortable when peers encouraged them to "be free" by swimming nude, as the participant was experiencing intense chest dysphoria and felt that their peers did not grasp their distress.

### 3.1.5. Category: Consequences of Identity Disclosure Decisions

In the aftermath of choosing to disclose or not disclose their identities, participants described the impacts of those decisions. One participant explained that their identity as a nonbinary person felt more validated in their host country, so they did not experience a significant adjustment period. While some participants chose to disclose their gender identities immediately upon arrival in their host country, others postponed the process of inviting people in. In one such instance, a participant's nonbinary identity became more prominent over the course of their time abroad, until it became something they wanted to "put forth in the world." Participants typically shared positive consequences of disclosing their identities, including feeling more authentic, comfortable, and connected to others; creating safer spaces for others to express themselves; and experiencing deep fulfillment in professional and leisure settings. One participant shared how disclosing their identity led them to rightfully take up space with other TGE students:

> *From the very first introduction with my study abroad program, everyone knew that I was queer. . . if you were queer, it was, we were all very open about it. In those introductions everyone was like " . . . my pronouns are they/them." So it was kind of like we were . . . making sure that our presence was known, and would not be forgotten or diminished.* (Participant 14)

In contrast, participants variantly described negative impacts that came with disclosing their gender and sexual identities. These impacts occurred along a spectrum that included instances of discomfort, annoyance, concerns about safety, and alienation. In many instances, participants categorized their experiences of coming out in the U.S. as negative, noting feelings of stress and lack of support. One participant discussed that being rejected by a friend after disclosing their TGE identity had motivated them to disclose their identity prior to establishing meaningful connections in the future to avoid similar situations. Another participant described a related experience in which the disclosure of their gender identity in the U.S. led to harm in an important familial relationship:

> *I came out to my dad as well who is Mexican . . . and he definitely was not okay with it and he's still not really okay with it. He still deadnames me and everything, uses he/him pronouns with me and stuff like that . . . I think one day, I'll stop talking to him because he can't accept my identity.* (Participant 3)

In addition to describing the positive and negative consequences associated with disclosing their identities to others, participants also discussed the positive and negative impacts resulting from choosing not to disclose their gender identities. Participants who did not disclose their gender to others variantly experienced negative consequences, including feeling unable to advocate for themselves, feeling less safe and connected, and feeling inauthentic. For example, a participant shared that they concealed their TGE identity from a romantic partner due to fearing his response, ultimately limiting the relationship's intimacy.

Conversely, participants variantly discussed positive consequences of not disclosing their gender identities, including feeling safer. For example, one participant explained that, while unfortunate, sometimes they have to conceal their TGE identity to keep themselves safe:

> *And sometimes . . . I would just have to closet myself because I knew that there are some battles that just can't be won about being truly my authentic self. And unfortunately, if this kind of fluidity of gender and presentation that I have can work sometimes to my favor in order to pass, then sometimes it has to work like that.* (Participant 7)

*3.2. Developing Relationships, Discerning Community, and Interacting with Others Domain*

All participants discussed the importance of community while abroad, and in this domain, participants explored the processes of creating friend, mentor, romantic/sexual, and other relationships. Participants described joy and affirmation gleaned through meaningful relationships abroad. One participant shared the intensely meaningful experience of having access to queer community abroad:

> *[Community] made the experience so much more meaningful because I felt seen. I felt present. I was in a foreign country . . . but I was seeing queer people that do call this place home living happily and enjoying my company. And I enjoyed theirs. . . I like to think about my study abroad experience as . . . this queer adventure because . . . I was always doing something with my queer friends. It felt like we were just this tight-knit family . . .* (Participant 14)

Participants also detailed the unfortunate realities of isolation, abuse, and assault that they experienced in community and interpersonal settings. Additionally, they illustrated their experiences with in/exclusion and access to communities that accepted and reflected their salient identities. Participants further discussed ways their universities helped or hindered their relationships and the significance of connection as part of learning and growth. Four categories of themes were discerned in the data. See Table 3 for a list of categories, subcategories, and frequencies.

**Table 3.** Categories and subcategories within the Developing Relationships, Discerning Community, and Interacting with Others domain.

| Categories (Frequencies) | Subcategories (Frequencies) |
|---|---|
| Energy investment in relationships (General, 15 participants endorsed) | Investing energy in relationships. (General, 14 participants endorsed) <br> Setting boundaries to limit relationships. (Typical, 11) <br> Seeking support/relationships but not finding it. (Typical, 9) |
| Outcomes of relationships or lack of relationships (General, 15) | Led to support and affirmation from others. (General, 15) <br> Led to unpleasant and harmful experiences. (General, 14) <br> Led to joyful experiences. (Typical, 13) |
| Ways of interacting with others (General, 15) | Academic/program-related. (Typical, 13) <br> Leisure. (Typical, 13) <br> Romantic and sexual interactions. (Typical, 13) <br> Living situation. (Typical, 8) <br> Political engagement and social justice activities. (Variant, 5) |
| Unique considerations in relationships (General, 15) | Disclosure. (General, 15) <br> Shared identities. (General, 15) <br> Intersectionality. (Typical, 11) <br> Brevity of study abroad. (Variant, 5) <br> Family of origin. (Variant, 2) |

3.2.1. Category: Energy Investment in Relationships

In this general category, all participants made decisions about how much energy they should invest in various relationships. Participants described the intentional acts of meeting and getting to know people. They also discussed the process of finding, creating, growing, and building communities abroad, particularly LGBTQIA+ and racial/ethnic identity communities. Each of these experiences required participants to intentionally invest energy in connecting with others. Many described engaging in intentional energy investment to build and maintain relationships, such as going out of their way to meet new friends abroad or educating others within relationships about aspects of their identity to cultivate a richer and more authentic interpersonal connection. One participant shared that, during study abroad, they chose to "show up" for someone they had known previously:

> *We were friends our freshman year and we stopped being friends . . . So we're going to South Africa as juniors, and we're like hey, I know we haven't really talked but we are going to a whole other country for the first time so if you need anything, I will be there. And I think, that just speaks to solidarity and actually putting solidarity in practice.* (Participant 12)

Typically, participants described engaging in boundary setting to increase their comfort with what occurred in relationships. They reported carefully selecting the situations and people in which they would invest energy and setting boundaries to reduce the energy they invested in interactions where it was not warranted or deserved. Participants described limiting what they shared with others and deciding who could and could not have continued access to their time and energy, usually based on how they had been treated previously. For example, a participant explained that they experienced misgendering as an act of violence, so they set firm boundaries that their close friends and lovers abroad could not repeatedly misgender them or the relationship would have to end.

Finally, participants typically discussed seeking out relationships and social support abroad, but not always finding what they were looking for. Participants expressed seeking and struggling to locate TGE, queer, Latine, and Black community while abroad. One

person who did not have relationships with other TGE people during study abroad reported that such relationships might have helped them feel less alone. In addition to struggling to access identity-related community, some participants also described feeling isolated from their study abroad cohorts and not developing meaningful relationships during study abroad.

3.2.2. Category: Outcomes of Relationships or Lack of Relationships

In this general category, participants described the experiences that resulted from their relationships and/or lack of relationships during study abroad. Generally, all participants discussed how relationships were critical to them feeling seen, supported, affirmed, important, wanted, appreciated, safe, and understood in their identities. Instances in which relationships led to support and affirmation included others sticking up and advocating for participants, accepting their identities "without question" and "with wide arms", and affirming the seriousness of and supporting them through experiences of assault and discrimination. Participants described emotional, logistical, nonverbal, and even physical support from others. One participant felt supported during a hike when they experienced mobility issues:

> *. . .there were people there who were willing to help me, they were being very patient and very understanding. I had a friend who walked with me, I had, the wife of the guy who was coordinating all of our events. . . his wife stayed right by my side the entire time . . . one of the people . . . had to carry me across the river. . . But just having, like, that sense of, these people are here for you, they're going to support you, they're going to make sure that you're going to get through this, you're going to be able to do it . . . was . . . probably one of the best things that has happened to me.* (Participant 4)

However, generally participants also reported interactions with other people that were unpleasant and harmful, many—but not all—of which were related to participants' bodies and identities, as discussed previously. Participants described experiences of intimate partner violence, transphobia, fatphobia, racism, sexual assault, verbal and physical harassment, misgendering, interpersonal challenges with professors and other students, discrimination in public spaces, slurs used against them, stares, tokenization, infantilization, lack of engagement, and entitlement. They reported feeling isolated, left out, excluded, lonely, disconnected, hurt, angry, unsafe, and tired as a result of these experiences. For example, a participant reported that locals in their host city felt entitled to ask invasive questions, which was exhausting:

> *It happens every time I met somebody in the local community and they're questioning my identity and questioning my name. I just really hate those conversations. Why can't they just accept when I say my name is [name]? They ask me "What's it mean?", "Is that a female name or a male name?", Or "Why do you dress this way?", "Were you born male or female?". . . It's like I have to justify my existence.* (Participant 15)

Typically, participants also reported joyful experiences with others during study abroad. They shared examples of having fun, celebrating, having great conversations, feeling alive and deeply connected, falling in love, meeting amazing people, expanding their world through new relationships, having "heartwarming" friendship experiences, feeling hopeful for "queer and trans futures", and experiencing profound solidarity. One participant described their experience of friendship during study abroad as an "explosion of amazement". Another shared that meeting others who were interested in talking and learning together was "beautiful". Finally, a participant who shadowed a gender affirming medical provider described an experience of joy and hope:

> *. . .the best part of that day in particular was being in the clinic and seeing a patient and that patient also had their supportive family. . . Seeing that and knowing that even though I might not have such a supportive family in the US like that, and knowing that that kind of affirmation and love can still happen in other places is just really. . . It just gave me hope.* (Participant 7)

### 3.2.3. Category: Ways of Interacting with Others

In this general category, all participants described the settings in which relationships occurred and the types of activities they completed with friends, chosen family, partners, host families, and other members of their community. They described a wide variety of interactions with other people while abroad. As they were all on academic study abroad programs, participants typically discussed interpersonal experiences within academic and program-related settings, including interacting and building relationships with other American study abroad students, international students from other countries, classmates, LGBTQIA+ student organizations, study abroad alumni, professionals working in their field of interest, faculty, study abroad administrators, and residential staff. One participant shared that they felt "extremely" supported by and connected to a professor abroad and felt comfortable being open with their gender identity and expression in the classroom because of this relationship. Describing this bond, the participant expressed relief: "Oh my god, you understand, thank you".

Typically, participants described interactions occurring within leisure settings; across participants, they noted camping, hiking, cooking potluck dinners, doing artistic activities, attending pride events, getting a tattoo, having sleepovers with friends, swimming in a lake, and going on excursions with friends. Many participants spoke about seeking out nightlife and clubs, where they had both positive and problematic experiences. One participant described experiencing sexual harassment in a night club abroad: "My friend and I went to a club ... and people tried to grab my face and force me to kiss them, and tried to grab my ass, so ... that was not a fun experience".

Typically, participants described experiences with romantic and sexual partners during study abroad. They shared a wide range of both positive and negative interactions, including using dating apps, going on dates, spending time with partners at bars and nightclubs, experiencing intimate partner violence and sexual assault, having consensual sexual encounters, studying abroad with a preexisting partner, and enduring breakups abroad. One participant shared that their romantic relationship ended during study abroad because their partner did not accept their gender and sexual identities.

Participants typically described the relationships and connections they built with those in their living environment. In their demographic surveys, participants reported primarily living with other students or with a host family. One participant felt connected to and seen by the other international students they lived with while abroad, as they were more politically liberal and accepting of trans identities than many of the locals. However, another participant, living in sorority-style housing with a large group of cisgender women, sometimes felt unsafe at home because others often misgendered or tokenized them.

Finally, participants variantly shared experiences of participating in political engagement and social justice activities while studying abroad, including taking part in protests and university elections. Feelings, including stress, excitement, and passion, often accompanied these experiences. One expressed a sense of LGBTQIA+ solidarity while protesting:

> *I think those places and protests ... with other queer and trans people have been incredibly empowering because I just saw them not give a shit about what people thought of them ... it was just a lot of mix of joy, resilience, fighting, being tired at the system but also realizing that sometimes we are the only ones looking out for each other.* (Participant 7)

### 3.2.4. Category: Unique Considerations in Relationships

In this general category, participants described elements of their relationships, interactions, and relational decision-making that were unique to their experiences of studying abroad as TGE students. Generally, disclosure was a unique relational consideration for participants, as trust and support in relationships often impacted participants' willingness to disclose their identities. Conversely, participants' decisions to disclose identity also impacted the level of trust and intimacy within their relationships. This disclosure subcategory does not explore the process of disclosure itself (as discussed earlier), but rather the ways disclosure impacted and was impacted by relationships. One participant discussed

that implicitly disclosing their TGE identity by wearing a trans flag button granted them access to several meaningful relationships abroad:

> ...*he asked me like, he was like, "Oh, can I ask you, are you like a trans ally or do you have a trans identity?"... I realized I had a button with a trans flag on it on my backpack... I was like, "Yeah I'm transgender..." And then he had revealed to me that he was transgender and he...invited me to join a group... I got to meet a lot of other people who were gender expansive ... it was really cool ... everyone was just so welcoming and so open.* (Participant 3)

Generally, participants described the immediacy of connecting with other queer and TGE people who they encountered during study abroad. In these cases, shared identities created a foundation of understanding and acceptance that accelerated relationship-building. One participant described the ways that shared gender, racial, and sexual identities impacted their relationships abroad compared to their relationships in the U.S.:

> *I didn't meet a lot of trans and nonbinary people and the ones that I did meet were White... I don't experience community with White people, in general. But I did meet a lot of Black queer people, Black lesbians, Black bisexual people in South Africa and that was helpful for me... I usually don't find community with cis people either. And when I'm [in the US], my friends are Black trans people ... so to go to South Africa and my closest friends were queer people who were cis was different for me.* (Participant 12)

As demonstrated in the previous participant quote, intersectionality was a typical interpersonal consideration for participants, highlighting dimensions of participants' intersectional identities that played a role in relationships and interactions. One participant described that, while other queer nightlife spaces centered the most privileged facets of the LGBTQIA+ community like White and cisgender people, they were excited to discover a nightclub abroad that celebrated their intersecting marginalized identities as a queer person of color.

Relationship-building abroad was variantly impacted by the brevity of study abroad, as most participants' educational programs lasted less than one year. Participants described the difficulty of cultivating long-term, sustainable friend and romantic/sexual relationships because of the time-limited nature of study abroad. One participant shared that, because they were only staying with their host family for a short time, it did not feel worth it to have honest conversations about identity, leading them to "tip toe" around the subject.

Finally, participants' families of origin variantly impacted their relationship-building experiences during study abroad. One participant reflected on how the transphobia they had experienced previously with their family of origin affected their ability to trust people and engage in relationships abroad. Another reported that a family member flew to their host country to help them settle in, which provided them with support during the first few weeks of study abroad.

## 4. Discussion

This study examined the experiences of 15 TGE adults from the United States who took part in an undergraduate study abroad program. Overall, exploring identity, reflecting on personal growth, and navigating relationships were significant aspects of their time abroad. In particular, their gender and other identities colored many aspects of their experiences, including their interactions with new cultures, their growth as students and people, and the relationship and community connections they developed while abroad.

### 4.1. Reflecting on Identity and Personal Growth

Participants spent considerable time and energy thinking through whether they wanted to or could disclose their identities while abroad based on their physical expression, the levels of trust in their relationships, and language and cultural barriers. Recalling the work of Doyle (2023), public self-identification as TGE is considered a TGE identity development milestone. As reflected in findings from the USTS (James et al. 2016) indicating

that the majority of TGE people are out to some or most people in their lives, deciding whether to disclose their identities while abroad was often a complex, multi-step process that varied depending on the ways participants felt in specific relationships, rather than being an all-encompassing choice. This process was further complicated by the study abroad backdrop of navigating new languages and cultures.

While this study sought to explore TGE experiences of study abroad, every participant shared the ways that they navigated and prioritized their gender identities alongside and in tandem with their intersecting marginalized and privileged identities like race, sexuality, ability, body size, and more. Identity salience is the likelihood that a person's identity/ies will be solicited during particular interactions or situations (Stryker 1968). Participants discussed that the salience of their various identities shifted based on their immediate environments, with contextual factors like visibility and privilege bringing certain identities to the forefront at different times. Interestingly, while prior research shows that often an individual's most salient identities are those that are marginalized (Jones and McEwen 2000), around half of our participants discussed the salience of their socially privileged identities while abroad. Many participants contended with the fact that, because they had entered a new culture, the ways they understood their own privilege and marginalization shifted. For example, one participant reported that their gender and racial identities are often highly salient as a Black TGE person in the U.S. However, while studying abroad, they found that their American identity provided them with privilege in comparison to the non-American Black people with whom they were interacting; in this way, their privileged identity as a Black American person became more salient while studying abroad.

All participants described feeling changed by study abroad, with some noting identity-specific growth and others describing aspects of internal change such as developing independence and patience. Consistent with the findings from the pilot study (Michl et al. 2019), despite an array of painful experiences, all participants reported that they changed or grew in some way during their study abroad experiences, and interestingly, if given the opportunity, they would study abroad again. The current study's findings related to personal growth support previous study abroad research. According to Bryant and Soria (2015), having the opportunity to explore queer and trans identities can be a study abroad motivator for many LGBTQIA+ students. Additionally, other research has shown that study abroad helps students grow in their self-understanding, patience, flexibility, and cultural sensitivity (Cisneros-Donahue et al. 2012).

Unfortunately, almost all participants described their experiences of contending with anticipated harm, experiencing actual harm, or navigating experiences of gender dysphoria while abroad. These findings reflect participants' experiences of distal and proximal minority stressors (Meyer 2003; Testa et al. 2015). Even for those who did not experience emotional or physical harm related to their identities, they were burdened with the possibility of being harmed as TGE people. The fear of being harmed as a result of having a TGE identity is valid and even realistic considering that TGE people are disproportionately likely to experience discrimination, harassment, and violence in nearly every setting in the U.S. (James et al. 2016). The findings from this study demonstrate that the threat of mistreatment and violence is not specific to a U.S. context. Previous research on the prevalence of harm experienced by cisgender and transgender students abroad is not available, but with regards to sexual violence specifically, a recent article reported that within a sample of 2630 self-identified male and female students, one in five who studied abroad reported experiencing sexual assault (Pedersen et al. 2020). While difficult to compare proportionally due to the finite sample size and qualitative nature of the current investigation, the data suggest that violence and harm are prevalent for TGE people in the U.S., study abroad students in general, and TGE study abroad students specifically. This reflects a need for more targeted protective measures before and during study abroad programs for all study abroad students and TGE students in particular.

Gender dysphoria is regarded as both a proximal minority stressor experienced by many TGE individuals (Lindley and Galupo 2020) and a milestone in trans identity devel-

opment (Doyle 2023). Because gender dysphoria is often considered a critical component of trans lived experiences, the researchers were surprised that gender dysphoria was only mentioned by two study participants. While many participants described instances of others communicating that their gender identity was misaligned with their appearance or other aspects of who they are, which can be understood as instances of distal minority stress and interpersonal harm, only two participants described the internalization of these social interactions resulting in the proximal stressor of gender dysphoria. It is possible that gender minority resilience factors such as identity pride and community connectedness (Testa et al. 2015) allowed participants who experienced misgendering and invalidation from others to resist the internalization of these messages, leading to an absence of gender dysphoria. It is also possible that, while only two participants explicitly named their experiences of gender dysphoria, others had similar internal experiences but did not interpret or verbalize these experiences in the same way. As CQR requires that researchers continuously come back to participants' descriptions of their own experiences in order to avoid interpretation, we did not feel comfortable ascribing the experience of gender dysphoria to participants who did not explicitly state this term.

Finally, participants described the consequences of the choices they made related to disclosure. Nine people discussed positive impacts of disclosing, seven described negative impacts of disclosing, four discussed negative impacts of not disclosing, and two described positive impacts of not disclosing. Identity concealment, or choosing not to disclose a TGE identity, is considered a proximal minority stressor, as it often stems from experiences or expectations of harm related to being "out" as a TGE person and can result in intense psychological distress (Testa et al. 2015). Although we live in a society where TGE people face daily violence and disclosing a trans identity may result in harm or death, the dominant cultural narratives about trans identities view coming out as a prerequisite to being a "valid" and "legitimate" TGE person. While some TGE individuals find that identity disclosure is worth the risk as it may result in increased access to authenticity and community, others may choose not to disclose for a variety of reasons. As experienced by our study participants, there are a variety of positive and negative consequences that can accompany decisions to disclose or conceal TGE identities; these choices are nuanced and dependent on individual priorities and safety considerations.

### 4.2. Developing Relationships, Discerning Community, and Interacting with Others

Relationships can enrich our lives and increase our wellbeing by providing access to joy, connection, solidarity, love, and more, and participants in this study described making decisions about investing energy in relationships with others and developing important relationships. For TGE people specifically, community is a central protective factor, having the power to protect those who are experiencing gender minority stressors from developing significant distress and mental health challenges as a result (Meyer 2003; Testa et al. 2015). While studying abroad, TGE students can face a wide variety of distal minority stressors, such as harassment and violence related to their gender identities, in addition to common stressors that all students may experience, such as language barriers, transportation issues, financial difficulties, and more. Because of the wide range of challenges encountered during study abroad, having access to a supportive community can be even more critical for TGE individuals. While abroad, it may be useful for TGE students to connect with others who share their gendered experiences and who can help them debrief and process after a distressing or joyful experience related to identity has occurred (Siddiqui and Jessup-Anger 2020). Additionally, making connections with other TGE and LGBTQIA+ individuals during study abroad can contribute to global trans and queer solidarity (Hipple et al. 2020).

Participants discussed that their relationships provided them with support, affirmation, and joy while abroad—highlighting the importance of relationships not only in order to protect TGE people from psychological distress and negative mental health outcomes (Meyer 2003; Testa et al. 2015), but also to provide them access to fun, meaningful, validating, and happy experiences. Although many reaped the benefits of feeling connected to

others, all but one participant also reported harmful or unpleasant relational experiences while abroad—both with straight, cisgender people and with other queer and TGE people. Unfortunately, although access to community can protect TGE people from negative mental health outcomes, community and relationships—even with those who have shared identities—are also often the settings where distal stressors like discrimination, harassment, and violence take place. For this reason, community during study abroad can be understood as a double-edged sword for TGE students: places for joy, affirmation, and support and for emotional and physical harm. It is critical to keep this duality in mind so as to avoid oversimplifying the role of relationships in the lives of TGE study abroad students.

Participants reported that interactions and relationships occurred in a variety of settings and activities abroad, including academic/program-related, leisure, romantic/sexual, living situation, and political engagement and social justice activities. It is important to note that leisure activities, particularly nightlife settings, were often places where interpersonal harm and violence occurred for our participants. Interestingly, as there were no interview questions related to political engagement or social justice activities, it is worth recognizing that one third of participants reported participating in such activities. This supports previous research by Eagan et al. (2016) that found that, compared to cisgender college students, TGE college students had more experience with civic engagement prior to college and a higher motivation to pursue activities related to social change after graduating. Researchers have also pointed to advocacy, activism, and other efforts toward social change as ways of coping with marginalization, as they can promote resilience, empowerment, and group solidarity (Hagen et al. 2018).

All participants described that having shared identities was a significant aspect of developing relationships abroad, as this often resulted in accelerated intimacy. While many participants described the importance and sacredness of being in community with other TGE people during study abroad, not all participants had access to or even wanted to pursue relationships with other TGE people. For some participants, having shared identities besides gender, such as race, was more important to them.

Participants discussed intersectionality when describing relationships and interpersonal interactions. Meyer's (2003) minority stress theory has been expanded through the integration of intersectionality theory (Crenshaw 1989), concluding that people who hold multiple marginalized identities are more greatly exposed to the harmful effects of stigma, ostracization, and inequity (Cyrus 2017; McConnell et al. 2018). McConnell et al.'s (2018) study further suggested that, while access to the queer community appeared to mediate the harmful effects of stress and stigma in White sexual minority men, the influence was less impactful for sexual minority men of color. While research applying the multiple minority stress model to TGE people of color remains minimal, Singh and McKleroy (2011) interviewed transgender people of color regarding their ability to remain resilient following a traumatic event. The study identified that pride in their racial/ethnic and gender identity, connecting with activist TGE communities of color, and cultivating hope for the future, among other components, facilitated resilience.

Prior research reflects differences in the needs, desires, and anticipation of harm for TGE people with multiple marginalized identities, which was also expressed by participants in the current study. While one participant expressed gratitude over access to a community of trans people abroad and the positive affirmation they experienced, another explained that, as the only nonbinary person on the trip, having another trans person with them likely would not have made a difference due to their history with discrimination from binary trans people. Yet another participant shared how their racial, gender, and sexual identities intertwined to influence how they anticipated harm: "I'm a Black individual, so that was already, that was the bigger concern for me, was more so, how will I be safe as a Black person? As opposed to how will I be safe as a queer person? How will I be safe as a nonbinary person"?

*4.3. Implications*

The findings from this study have widespread implications for various stakeholders. Students, especially TGE and queer individuals, and their families may want to consider how these findings might inform their study abroad plans. Additionally, psychotherapists in college counseling and other settings can help students navigate and make sense of study abroad experiences during and after international programs; therefore, it is hoped that these results will help enrich their understanding of common experiences encountered by TGE students so they can be better supported. Finally, educational institutions and organizations around the world offer study abroad opportunities, often generating profits. It is organizations—including their leadership, faculty, and staff—that can implement structural improvements to enhance the wellbeing of all students, especially TGE students who may be more vulnerable due to societal discrimination. Therefore, recommendations are made here specifically to enhance study abroad programs.

4.3.1. Recommendations for Study Abroad Interventions

Findings from this study demonstrate several gaps in the study abroad experiences of TGE individuals that can be addressed by universities, organizations, and study abroad programs. First, as harm and violence were common occurrences in this study and previous research (Pedersen et al. 2020), programs need to develop and disseminate concrete measures to protect TGE students from emotional, physical, and sexual violence during study abroad, particularly in leisure settings like nightlife. For example, all study abroad students should receive education about common dangers, ways to protect themselves, and clear steps to take to seek help and report harm when necessary. Second, programs can stress that building relationships is often one of the most important and fulfilling aspects of study abroad, and thus provide resources to connect TGE individuals to community in their host countries. Programs should also bear in mind that many TGE students benefit from having access not only to LGBTQIA+ community while studying abroad, but also to other kinds of communities that align with their intersecting identities, such as communities oriented towards Black students or students with disabilities. A fundamental component in crafting study abroad programs that more effectively support TGE students is to recognize the need for more representation of marginalized identities within student cohorts, professors, and curriculum. Third, programs may want to connect TGE students with advocacy and activism opportunities in order to support them in building community and enhancing wellbeing while abroad. Finally, knowing that study abroad will likely be a time of immense personal growth, identity exploration, learning, relationship building, and possibly interpersonal harm, programs can develop or enhance opportunities for students to process their experiences. For instance, programs can build in reflective assignments or schedule meetings to discuss study abroad experiences individually or in small groups during their time away and after they return home. Additionally, programs should ensure that TGE students have clear instructions about how to access mental healthcare while abroad and upon their return to their home campus, should they need to discuss experiences with a therapist.

4.3.2. Recommendations for Future Research

This study illuminates many areas of possible future research, all of which can have important benefits for the wellbeing of TGE people. It is recommended that more research, especially mixed-methods research, be conducted on the experiences of TGE students who study abroad. This is critical, as the authors are not aware of other work outside of the pilot study (Michl et al. 2019) that specifically explores TGE study abroad experiences, yet the initial research suggests that LGBTQIA+ students may be more likely to study abroad (Bryant and Soria 2015). Similarly, with the increases in international students studying in the United States, research into how TGE students navigate living and studying in America or other popular host countries would be beneficial. Additionally, we recommend researchers examine the complicated, non-linear, and relationship-based decisions to dis-

close or conceal TGE identities during study abroad or even during international leisure travel. We also suggest that researchers explore the ways that TGE individuals make sense of their intersecting identities in a variety of cultural contexts, particularly attending to how privilege impacts salience when in a home versus host country. Additionally, in light of so many participants experiencing harm, it is hoped that future intervention research will clarify effective strategies in maintaining the physical safety of TGE students during study abroad, and longitudinal research will reveal the ways that harm and anticipated harm impacts TGE individuals over time. In terms of community-focused research, we recommend that scholars explore the "double-edged sword" of community—serving as both a gender minority stress and resilience factor—for TGE individuals in a variety of settings. Finally, as TGE students in this study defined "community" in many different ways based on their unique identities and needs, it is recommended that researchers continue to explore the ways TGE individuals identify and construct community.

*4.4. Limitations*

This study is qualitative and exploratory in nature and sought to deeply examine the stories of 15 TGE students to generate relevant themes about study abroad experiences. These findings are not intended to represent the experiences of all TGE study abroad students. Additionally, the current project is designed to understand TGE study abroad experiences in a variety of study abroad locations, rather than exploring cultural norms or gender considerations for one specific international location. That said, individuals who studied in European countries are overrepresented in our sample. Finally, the majority of study participants were in their early twenties; while this is a common age range among American undergraduate study abroad students, it may have been useful to include older and non-traditional students to incorporate perspectives of TGE individuals from a variety of generations.

**5. Conclusions**

Studying abroad can serve as a chance for college students to freely explore themselves and how they fit in the world. Shedding the baggage of familial expectations or the identity they have settled into over the course of a lifetime is a rare opportunity, made feasible by the physical distance and cultural departures between "home" and a host city. For TGE students, studying abroad has the potential to offer a safe, supportive gateway to unprecedented personal growth and connection to global community. The TGE study abroad students in this study invested a great deal of time and energy in negotiating identity- and community-related concerns. While gender was often an important identity, participants described the ways their many intersecting identities came into play during their time abroad. Participants made difficult decisions about disclosing or concealing their gender identities while abroad. Additionally, participants experienced the joys of community and the perils of interpersonal violence. Participants contended with their intersectional identities; grew as people; met friends and lovers; set boundaries in relationships; and navigated the complex interactions between identity, culture, and relationships during study abroad. Their experiences demonstrate that identity and relationship considerations are multifaceted when it comes to the study abroad experiences of American TGE students; these considerations warrant increased attention from researchers and study abroad programs to ensure the safety, wellbeing, and enrichment of these students.

**Author Contributions:** Conceptualization, T.M. and A.K.; data collection, T.M. and A.K.; data analysis, T.M., A.S., J.W., K.C. and A.K.; auditing, T.L.R.; writing—original draft preparation, T.M.; writing—review and editing, T.M., A.S., J.W., K.C., T.L.R. and A.K. All authors have read and agreed to the published version of the manuscript.

**Funding:** This research received no external funding.

**Institutional Review Board Statement:** The study was conducted in accordance with the Declaration of Helsinki, and approved by the Institutional Review Board of Webster University (SP20-44, 7 August 2020).

**Informed Consent Statement:** Informed consent was obtained from all subjects involved in the study.

**Data Availability Statement:** The data presented in this study are available on request from the corresponding author. The data are not publicly available to maintain the participants' privacy.

**Acknowledgments:** The authors would like to warmly thank all individuals who participated in this study. We appreciate Ander Shunneson's early contributions as a research team member, as well as support of Webster University, where we first started this project. The authors received no funding to support this work. An abbreviated version of these findings was presented at the 2023 World Conference of Qualitative Research in Faro, Portugal and the 2023 American Psychological Association Annual Convention in Washington, DC, USA.

**Conflicts of Interest:** The authors declare no conflict of interest.

## Appendix A. Semi-Structured Qualitative Interview Guide

Today I'm going to be asking you questions about your study abroad experiences. Can we first review where and when you studied abroad?

1. Can you briefly tell me about how and why you chose to study abroad?
   a. How did you choose your location?
2. How did you prepare for your trip?
   a. How did you feel before leaving to study abroad?
3. How would you compare your lived experience as a transgender or gender expansive person in the United States versus in your host country?
4. What meaningful relationships did you develop while abroad? This can include your housemate(s), friends, romantic/sexual partners, teachers, etc.
   a. How, if at all, were these relationships affected by your gender and other identities?
5. Did you have relationships (of any kind) with transgender and/or gender expansive people in your host country?
   a. How did this change/would this have changed your experience, if at all?
6. How did you navigate identity during study abroad?
   a. Which identities did you share with/disclose to others?
   b. Which identities were most prominent/salient for you?
7. In general, how would you compare the decision to "come out" and/or the process of "coming out" in the U.S. versus in your host country?
8. What was the best experience you had while studying abroad?
9. What was the worst experience you had while studying abroad?
10. What were your experiences with issues of safety while living abroad?
    a. What were your experiences with harassment and violence while abroad? This could be verbal, emotional, physical, or sexual in nature.
11. Have you changed or grown as a result of studying abroad? If so, how?
12. When you reflect back on study abroad, do you feel your identities affected your experiences? If so, how? This might include identities such as gender, race, religion, ability, sexuality, etc.
13. If you had the opportunity, would you study abroad again? Why or why not?
14. What advice would you give to other transgender and gender expansive students who are considering studying abroad?

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
