# Peer review of "Trans Abroad: American Transgender Students’ Experiences of Navigating Identity and Community While Studying Abroad"

_socsci, doi:10.3390/socsci12090472_

Round 1

Reviewer 1 Report

This is very well-written paper. Congratulations! I particularly enjoyed the implications section of the paper.

There are spacing issues on Table 1. I do not completely understand Tables 2 and 3. I do not know understand the repetition of some information between the 2 columns, and I do not understand what general, typical, and variant mean.

Author Response

We have copied each comment we received and provided our accompanying responses below. In addition, we have uploaded a new draft of the manuscript, with changes referenced in this letter highlighted in yellow.

Reviewer 1:

  1. This is very well-written paper. Congratulations! I particularly enjoyed the implications section of the paper.

Thank you so much for these kind words. We are thrilled that you enjoyed the paper and the implications section in particular. Since beginning this research, we have hoped that our work would meaningfully and positively impact future trans students’ study abroad experiences on an individual and systemic level. For this reason, we are excited to hear that the implications section stood out to you. 

  1. There are spacing issues on Table 1. I do not completely understand Tables 2 and 3. I do not know understand the repetition of some information between the 2 columns, and I do not understand what general, typical, and variant mean.

We appreciate this feedback related to the spacing and formatting of all tables. We have resolved the spacing issues in Table 1. Tables 2 and 3 mistakenly contained repetitive information previously. These errors have been removed and spacing issues have been resolved in both tables as well to ensure clarity. 

The consensual qualitative research (CQR) method requires that the frequencies of domains, categories, and subcategories are reported to demonstrate to what extent themes represent the experiences of the sample. For this reason, we have reported on the number of participants who endorsed each domain, category, and subcategory, and used the CQR frequency labels of General (14 or 15 participants endorsed), Typical (8-13 participants endorsed), and Variant (2-7 participants endorsed) to distinguish them, as is required for all CQR studies. Tables 2 and 3 now contain more specific labels (for example, “General, 15 participants endorsed”) to clarify the meaning of frequency labels. Frequencies are explained in the final paragraph of the Data Analytic Strategies section, as follows:

“CQR requires that the frequencies of domains, categories, and subcategories are reported to demonstrate to what extent themes represent the experiences of the sample (Hill & Knox 2021). Domains, categories, and subcategories that are endorsed by 14 or 15 participants are labeled General; 8-13 participants are labeled Typical; 2-7 participants are labeled Variant; and those endorsed by one participant are not reported. According to Hill and Knox (2021), the emergence of General and Typical categories is an additional indication that saturation has been reached.”

Reviewer 2 Report

The paper is very well written and timely, and I have just a few editorial suggestions prior to publication.

The title portion in quotes might be removed. It's not overly compelling and is a bit awkward to read. Titles with quotes in them also, according to several sources, are sourced less frequently than titles without them. 

Intro- references are required in the two opening pghs

Materials and methods- references in opening pgh. Although the study is described as qualitative, there is a survey involved and some of the findings seem to be influenced or organized in a way that could be read as quantitative (i.e., frequencies). Perhaps "mixed methods" is a more apt descriptor. 

The findings and discussion is fascinating and very soundly organized. 

Author Response

We have copied each comment we received and provided our accompanying responses below. In addition, we have uploaded a new draft of the manuscript, with changes referenced in this letter highlighted in yellow.

Reviewer 2:

  1. The paper is very well written and timely, and I have just a few editorial suggestions prior to publication.

Thank you so much for these kind words related to our writing and the timeliness of this work. Because transgender and gender expansive individuals are facing hostile and discriminatory legislation at this moment in time, it is our hope that our research will contribute to trans students experiencing safety, joy, and connection during study abroad. We hope that, against the backdrop of widespread transphobia, this work offers pathways to hope and resilience during study abroad.

  1. The title portion in quotes might be removed. It's not overly compelling and is a bit awkward to read. Titles with quotes in them also, according to several sources, are sourced less frequently than titles without them.

We appreciate this thoughtful feedback related to the manuscript’s title. After reviewing your suggestions, we agree that a title without quotes would be clearer and more compelling. We have changed the title from “Feeling empowered to just keep on existing”: American Transgender Students’ Experiences of Navigating Identity and Community While Studying Abroad to Trans Abroad: American Transgender Students’ Experiences of Navigating Identity and Community while Studying Abroad.

  1. Intro- references are required in the two opening pghs

Thank you so much for pointing out that the first two paragraphs of the introduction warranted additional references. We agree. The following references have been added to the first two paragraphs:

  • “As identities are socially and culturally constructed (Jackson & Hogg 2010), studying internationally offers students a rare opportunity to examine who they are against the backdrop of a new culture.”
  • “TGE people have gender identities that do not align with their sex assigned at birth (NCTE 2023).”
  • “Sadly, 19% of all reported hate crimes in the U.S. were targeted towards LGBTQIA+ communities in 2021 (FBI 2021), and at least 34 transgender and gender expansive individuals were murdered in the U.S. in 2022 (HRC 2022). According to the ACLU (2023), there are currently 492 active anti-LGBTQIA+ bills in the U.S. It is widely recognized that such legislation causes harm.”

  1. Materials and methods- references in opening pgh. Although the study is described as qualitative, there is a survey involved and some of the findings seem to be influenced or organized in a way that could be read as quantitative (i.e., frequencies). Perhaps "mixed methods" is a more apt descriptor.

Thank you for bringing attention to the need for more references in the opening paragraph of Materials and Methods. Two additional in-text citations have been added:

  • “CQR is particularly useful for exploratory inquiry of topics that have not been extensively studied (Hill & Knox 2021).”
  • “Additionally, CQR acknowledges the inevitability of researcher biases, invites open conversation about these biases, and uses multiple perspectives, consensus, and auditing to bracket biases and minimize their impact on data collection and analysis (Hill & Knox 2021).”

Thank you also for your feedback about how our methodology is described. In your comment, you share that, due to the demographics survey participants were asked to fill out and our use of frequencies to label our qualitative findings, you wonder if mixed method may be a more apt descriptor of our approach. The consensual qualitative research (CQR) method, which is widely used in psychology, requires that the frequencies of domains, categories, and subcategories are reported to demonstrate to what extent themes represent the experiences of the sample. For this reason, we have reported on the number of participants who endorsed each domain, category, and subcategory, and used the CQR frequency labels of General (14 or 15 participants endorsed), Typical (8-13 participants endorsed), and Variant (2-7 participants endorsed) to distinguish them, as is required for all CQR studies. Tables 2 and 3 now contain more specific labels (“General, 15 participants endorsed”) to clarify the meaning of frequency labels. Frequencies are explained in the final paragraph of the Data Analytic Strategies section, as follows:

“CQR requires that the frequencies of domains, categories, and subcategories are reported to demonstrate to what extent themes represent the experiences of the sample (Hill & Knox 2021). Domains, categories, and subcategories that are endorsed by 14 or 15 participants are labeled General; 8-13 participants are labeled Typical; 2-7 participants are labeled Variant; and those endorsed by one participant are not reported. According to Hill and Knox (2021), the emergence of General and Typical categories is an additional indication that saturation has been reached.”

 Additionally, as is standard for qualitative research, some basic demographics were collected from participants. While we would love for future research to explore the study abroad experiences of TGE students using mixed methods approaches, unfortunately the current study does not meet the rigorous requirements to be considered a mixed methods study.

  1. The findings and discussion is fascinating and very soundly organized.

We are very appreciative of this positive feedback related to the findings and discussion section. Because the study abroad experiences of TGE students can be so complex, it is our hope that the findings and discussion sections of this paper both honor this nuance while also presenting information in an accessible manner.

Reviewer 3 Report

Thank you all for this submission. There is, as you've rightly noted, a serious lack of quality research on the experiences of gender nonconforming folk in relation to studying abroad, and I feel that this paper is a valuable contribution to the field. 

As the authors rightly point out, this study has a small sample size, but that's not uncommon with the type of qualitative research they've chosen to employ. 

A small suggestion however: in section 2.5.1 (line 265), I think it may be helpful to discuss if this study's sample size allowed researchers to reach theoretical saturation, or if a larger sample size could have provided additional insights. 

I'm also interested in line 322, specifically the decision not to use qualitative software. What was the rationale behind this decision? I have no real reservations about it, but I am curious (and I imagine other readers might be as well) as to why the decision was made. 

Author Response

We have copied each comment we received and provided our accompanying responses below. In addition, we have uploaded a new draft of the manuscript, with changes referenced in this letter highlighted in yellow.

Reviewer 3:

  1. Thank you all for this submission. There is, as you've rightly noted, a serious lack of quality research on the experiences of gender nonconforming folk in relation to studying abroad, and I feel that this paper is a valuable contribution to the field.

Thank you so much for this comment; we appreciate that you see our work as a valuable contribution to the field. It is unfortunate that there is such a huge gap in the literature related to trans study abroad experiences. We feel lucky to have the resources to explore the complexities of these experiences, and hope that this paper positively impacts both individual trans study abroad students and the institutions that they contend with during study abroad.

  1. As the authors rightly point out, this study has a small sample size, but that's not uncommon with the type of qualitative research they've chosen to employ.

We appreciate and agree with this statement, that a sample size of 15 participants is not uncommon for qualitative research and, more specifically, the consensual qualitative research (CQR) method.

  1. A small suggestion however: in section 2.5.1 (line 265), I think it may be helpful to discuss if this study's sample size allowed researchers to reach theoretical saturation, or if a larger sample size could have provided additional insights.

Thank you so much for this thoughtful suggestion to confirm that the study’s sample size allowed for theoretical saturation to be achieved. We have added three statements in the Materials and Methods section to clarify this:

  • Section 2.4: “It is recommended that CQR studies use 13-15 participants to yield consistency across participant narratives and achieve saturation (Hill & Knox 2021).”
  • Section 2.6.2: “Whereas each participant shared unique experiences, several patterns and themes emerged throughout data analysis. In line with Hill and Knox (2021), the majority of new categories and subcategories emerged while analyzing the first two thirds of cases, indicating that saturation had been reached.”
  • Section 2.6.2: “According to Hill and Knox (2021), the emergence of General and Typical categories is an additional indication that saturation has been reached.”

  1. I'm also interested in line 322, specifically the decision not to use qualitative software. What was the rationale behind this decision? I have no real reservations about it, but I am curious (and I imagine other readers might be as well) as to why the decision was made.

Thank you for this feedback related to our choice not to use qualitative software. Not using qualitative software is standard practice in the consensual qualitative research (CQR) method; this approach allowed our research team to remain close to the data throughout the duration of the data analysis process. The following statement has been added to section 2.6.2: “Qualitative software was not used to organize data, as is standard in CQR; this approach allowed the researchers to remain close to the data throughout the duration of analysis.”

Reviewer 4 Report

The article is pertinent and focuses on a topic that is still under-researched, that of trans students studying abroad.

The methodology is clear and adequate, and the ethical issues (including the position of the persons who authored the article) are quite pertinent and well developed in the article.

Regarding the presentation of the results, it seems to us that the tables could be improved. Table 1 needs editing for formatting and, in addition, we do not believe that the data should be presented in percentages. Since the "n" is only 15, there is no risk of losing the notion of proportions. With regard to Table 2, we consider that the rationale for the two columns should be explained.

Regarding the results, although this is an exploratory research, involving only 15 participants, it produces interesting results that could be further expanded in future research.

Author Response

We have copied each comment we received and provided our accompanying responses below. In addition, we have uploaded a new draft of the manuscript, with changes referenced in this letter highlighted in yellow.

Reviewer 4:

  1. The article is pertinent and focuses on a topic that is still under-researched, that of trans students studying abroad.

Thank you so much for this comment; we are delighted that you see the research as pertinent and filling a gap in the research literature related to trans study abroad. We hope that results from this study make tangible impacts on future trans study abroad students and the institutions they interact with throughout their studies.

  1. The methodology is clear and adequate, and the ethical issues (including the position of the persons who authored the article) are quite pertinent and well developed in the article.

We very much appreciate this feedback. It is helpful to know that the methodology, ethical issues, and positionality portions of the manuscript feel well-developed to you. We engaged in several conversations about these aspects of the study throughout data collection and analysis, and it is incredibly important to our team to do ethical research with marginalized communities like trans and gender expansive folks.

  1. Regarding the presentation of the results, it seems to us that the tables could be improved. Table 1 needs editing for formatting and, in addition, we do not believe that the data should be presented in percentages. Since the "n" is only 15, there is no risk of losing the notion of proportions. With regard to Table 2, we consider that the rationale for the two columns should be explained.

We appreciate this feedback related to the spacing and formatting of all tables. We have resolved the spacing issues in Table 1 and removed percentages. Tables 2 and 3 mistakenly contained repetitive information previously. These errors have been removed and spacing issues have been resolved in both tables as well to ensure clarity.

  1. Regarding the results, although this is an exploratory research, involving only 15 participants, it produces interesting results that could be further expanded in future research.

Thank you so much for this comment. We are delighted that you found the results interesting and we hope that findings from this study motivate future research that clarifies additional components of trans students’ study abroad experiences.